Quality of life and subjective well-being comparison between traumatic, nontraumatic chronic spinal cord injury, and healthy individuals in China

Altahla Ruba 1
Alshorman Jamal 2
Ali-Shah Sayed Zulfiqar 3
http://orcid.org/0000-0002-0617-9035 Nasb Mohammad 4
Tao Xu 1 I202122089@hust.edu.cn
1 Department of Rehabilitation, Tongji Medical College, Huazhong University of Science and Technology , Wuhan, Hubei , China
2 Orthopedic Department, Xianning Medical College, The Second Affiliated Hospital of Hubei University of Science and Technology , Xianning, Hubei , China
3 Head of Rehabilitation, TopSupport International Sports Performance and Rehabilitation Center, Qingzhen Sports training base , Guizhou, Hubei , China
4 Tianjiu Research and Development Center for Exercise Nutrition and Foods, Hubei Key Laboratory of Exercise Training and Monitoring, College of Sports Medicine, Wuhan Sports University , Wuhan, Hubei , China
Teh Cindy Shuan Ju
Electronic publication date: 2024 Dec 23
Publication date: 2024
Volume: 12
Electronic Location ID: e18709
Received 2023 May 17; Accepted 2024 Nov 23
Copyright: © 2024 Altahla et al.
Copyright year: 2024
Copyright holder: Altahla et al.
License: This is an open access article distributed under the terms of the Creative Commons Attribution License, which permits unrestricted use, distribution, reproduction and adaptation in any medium and for any purpose provided that it is properly attributed. For attribution, the original author(s), title, publication source (PeerJ) and either DOI or URL of the article must be cited.
License URL: https://creativecommons.org/licenses/by/4.0/

Keywords: Quality of life, Subjective well-being, Satisfaction with life, Traumatic spinal cord injury, Nontraumatic spinal cord injury

Funding: National Natural Science Foundation of China 81772440 This work was supported by the National Natural Science Foundation of China (Grant No. 81772440). The funders had no role in study design, data collection and analysis, decision to publish, or preparation of the manuscript.

==============================
Background

Differentiating between traumatic and non-traumatic spinal cord injuries (NT-SCI) is critical, as these classifications may significantly impact patients’ health outcomes and overall well-being, potentially resulting in differences in treatment protocols and therapeutic efficacy.

Objective

This study aims to compare the quality of life (QoL) and satisfaction with life (SWL) among individuals with traumatic spinal cord injuries (T-SCI), NT-SCI, and the healthy population in China.

Method

A quantitative, cross-sectional survey was conducted between July and December 2020 in the Rehabilitation and Physiotherapy Department of Tongji Hospital, Hubei University of Science and Technology. The World Health Organization Quality of Life Brief Version (WHOQOL-BREF) and the Satisfaction with Life Scale (SWLS) were administered to evaluate QoL and subjective well-being (SWB). An independent t-test was performed to assess differences within the SCI population, while Pearson’s correlation coefficient was utilized to explore relationships between the WHOQOL-BREF domains and the SWLS. Multiple linear regression analysis was applied to identify key determinants influencing World Health Organization Quality of Life (WHOQOL) domain scores and overall SWLS score.

Result

Participants with NT-SCI exhibited significant differences in overall health as measured by the WHOQOL-BREF domains compared to those with T-SCI (p < 0.05). Both the NT-SCI and T-SCI groups demonstrated lower mean scores across all four WHOQOL-BREF domains compared to healthy individuals. No significant differences were observed between the NT-SCI and T-SCI groups in the SWLS, although both groups scored lower than the healthy population. Additionally, SWB was moderately positively correlated with QoL (p < 0.01). Collectively, the predictor variables explained 22.28% of the variance in physical health, 6.43% in psychological health, 28.67% in social health, and 25.68% in environmental health. Furthermore, the independent variables accounted for approximately 22.67% of the variance in the SWLS.

Conclusions

Individuals with NT-SCI experience significantly worse overall health outcomes compared to those with T-SCI, although both groups report QoL and life satisfaction than healthy individuals. No significant differences in life satisfaction were found between two groups. Additionally, SWB shows a moderate positive correlation with QoL, highlighting the close relationship between mental and physical health in SCI populations.

Introduction

Spinal cord injury (SCI) is categorized by etiology into traumatic spinal cord injury (T-SCI) and non-traumatic spinal cord injury (NT-SCI) or dysfunction which results from damage to the spinal cord tissue (Molinares et al., 2022; Altahla et al., 2024). The global incidence rate of T-SCI is estimated to range between 16 and 40 cases per million individuals annually (Chhabra et al., 2018). In contrast, the incidence of NT-SCI tends to be higher in developed nations compared to developing countries, likely due to more systematic and accurate reporting (New, Simmonds & Stevermuer, 2011; Guilcher et al., 2015). Demographically, most T-SCI cases involve younger, single males, whereas NT-SCI predominantly affects older, married individuals, with a more equal distribution between men and women (Molinares et al., 2022; Lee et al., 2014). Given these demographic and etiological differences, disparities in quality of life (QoL) and subjective well-being (SWB) between these groups may be anticipated. Several published studies have examined the functional outcomes, secondary health conditions, QoL, and satisfaction with life (SWL) in individual with T-SCI (Barclay et al., 2017; Nizeyimana, Joseph & Phillips, 2022; New et al., 2017; Chang et al., 2020). Studies have predominantly focused on T-SCI, with most SCI studies centered on this group (New et al., 2017; Chang et al., 2020; Altahla, Alshorman & Tao, 2023). In contrast, limited research addresses QoL and SWL among individuals with NT-SCI, where only a few studies have included this population (Barclay et al., 2017; New et al., 2017). Moreover, existing studies have not differentiated between various QoL and SWL factors. Chang et al. (2020) utilized the World Health Organization Quality of Life-Brief Version (WHOQOL-BREF) for T-SCI QoL assessments, supplementing this with the WHO Disability Assessment Schedule (WHOQOL-DIS) modules to better capture disability-related dimensions. In China, individuals with chronic SCI exhibit lower QoL compared to the general population, underscoring the need for detailed assessments, particularly in NT-SCI, during rehabilitation and community reintegration. Recent research highlights the importance of addressing QoL and SWB in SCI populations, integrating health and well-being as central components of QoL assessments. Notably, one of the proposed future aims of the International Classification of Functioning (ICF) is to strengthen “links with QoL concepts and the evaluation of subjective well-being” (Chang et al., 2012).

To date, the literature reveals a gap in studies specifically addressing QoL and SWB in NT-SCI populations in China. Therefore, this study aims to compare QoL and SWL among Chinese individuals with T-SCI, NT-SCI, and a healthy control group, as well as to assess the association between QoL and SWB in adults with SCI.

Materials and Methods

Participants

This quantitative, cross-sectional survey was conducted in the Rehabilitation and Physiotherapy Department of Tongji Hospital, China among individuals with SCI from July to December 2020. Convenience sampling was used to recruit participants exclusively from Tongji Hospital, and all questionnaires were administered in a written format with checkbox responses. For participants unable to complete the questionnaires independently, three trained researchers provided assistance, while self-administration was encouraged whenever possible. Inclusion criteria required participants to be at least 18 years old, have traumatic or non-traumatic SCI for a duration exceeding 6 months, and reside within the community. Individuals with mental health or cognitive impairments were excluded from the study.

The survey instrument

The survey included demographic and injury-related questions, alongside two validated instruments: the WHOQOL-BREF for assessing quality of life and the Satisfaction with Life Scale (SWLS) (Hao, 2000; Diener et al., 1985).

Quality of life

The WHOQOL-BREF scale was utilized to assess QoL among individuals with physical disabilities (Power, Green & Group, 2010). Introduced in mainland China in 1998 (Hao, 2000), the Chinese version of the WHOQOL-BREF has demonstrated acceptable reliability, validity, and suitability as a comprehensive generic health-related QoL measure, supporting its use in studies of QoL and disability among Chinese individuals with SCI (Lin et al., 2007; Tian, 2009). The initial two items of the Chinese version assess general QoL and overall health. The remaining 24 items are rated on a five-point Likert scale and categorized into four domains: physical health (seven items), psychological well-being (six items), social relationships (three items), and environmental health (eight items). Domain-specific summary scores were calculated and converted to a 0–100 scale, following the manual’s guidelines, where a score of 0 denotes the lowest QoL and 100 signifies the highest QoL (World Health Organization, 1998). In this study, after conducting the initial assessment, the WHOQOL-BREF questionnaire was re-administered to 30 participants after a 2-week interval. The Intraclass Correlation Coefficient (ICC) was then computed to evaluate the reliability of the measurements. The Cronbach’s alpha for the WHOQOL-BREF was found to be 0.82, indicating good internal consistency, while the ICC for the total scale score was 0.95, with subscale ICC values ranging from 0.89 to 0.98, demonstrating excellent reliability.

The satisfaction with life scale (SWLS)

The Satisfaction with Life Scale (SWLS) is one of the most widely used instruments for measuring subjective well-being (Diener et al., 1985). The SWLS demonstrates validated psychometric properties, including high internal consistency and acceptable test-retest correlations (López-Ortega, Torres-Castro & Rosas-Carrasco, 2016). Developed in 1985, the SWLS was specifically designed to assess subjective well-being across various patient populations, including individuals with spinal cord injuries (Post et al., 2012). The scale comprises five items, each rated on a seven-point Likert scale (e.g., Item 1: “In most ways, my life is close to ideal”). All five items are positively keyed, allowing for the calculation of an overall score. Scores range from 5 to 9, indicating “extremely dissatisfied” with life; 20 to 30, signifying a “good” level of satisfaction; and 31 to 35, reflecting “extremely satisfied” respondents, with scores in this range suggesting a high level of life satisfaction (Pavot & Diener, 2008). In this study, we analyzed scores from 223 Chinese participants aged over 18 years and compared them to those of healthy Chinese adults recruited from the general community. Following the initial assessment, the WHOQOL-BREF questionnaire was re-administered to 30 participants after a 2-week interval. The Intraclass Correlation Coefficient (ICC) was calculated to measure reliability. The Cronbach’s alpha for the scale was 0.89, and the ICC for the total scale score was 0.98, indicating excellent reliability.

Statistical analyses

Data analysis was conducted using SPSS version 23 and R (version 4.0.2, R Foundation for Statistical Computing, Vienna, Austria), reporting descriptive statistics, means, and standard deviations. The data were found to follow a normal distribution, and Student’s independent t-test was computed to explore differences between SCI groups. Additionally, Pearson’s correlation coefficients were calculated to determine associations between the WHOQOL-BREF and the SWLS.

Multiple linear regressions were utilized to identify key determinants of the WHOQOL domain scores, excluding items 1 and 2, with the SWLS score serving as the outcome variable. Variables were selected based on their theoretical or clinical significance, as controls to mitigate confounding effects, and according to their statistical significance. This methodical approach aimed to ensure the robustness of the model by integrating essential clinical insights and statistically reliable relationships.

Cronbach’s alpha was calculated to evaluate the internal consistency of the WHOQOL-BREF questionnaire, while test-retest reliability was assessed through the application of the ICC. All statistical tests were two-tailed, with a significance level set at p < 0.05. The research project was approved by the ethics committees of two affiliated hospitals of Tongji Medical College, with ethical approval granted under number TJ-IRB20210314.

Results

Participant characteristics

A total of 189 individuals with SCI were included in the study. Among these patients, 121 (64.02%) had T-SCI, while 68 had NT-SCI. The NT-SCI group included 17 (25%) patients with triple aortic aneurysms, 14 (20.58%) with transverse myelitis, 11 (16.17%) with neoplasm, 13 (19.11%) with infections/abscesses, 6 (8.82%) with viral infections, and seven (10.29%) with canal stenosis. The primary causes of T-SCI were road traffic accidents in 41 (33.88%) patients, falls in 38 (31.40%) patients, other accidents in 13 (10.74%) patients, work-related injuries in 20 (16.52%) patients, and sports-related injuries in nine (7.43%) patients. The most of T-SCI group was with level A 39 (32.23%) of neurological severity, while 38 (55.88%) with D neurological severity in NT-SCI group (see Table 1).

Table 1 Sample demographics.

Characteristic	Category	T-SCI
(n = 121)	NT-SCI
(n = 68)	
Level of injury (n%)	C1–C4	40 (33.05%)	5 (7.35%)	
C5–C8	26 (21.48%)	11 (16.17%)	
T1–T6	11 (9.09%)	10 (14.70%)	
T7–T12	36 (29.75%)	32 (47.05%)	
Lumbar or sacral	8 (6.61%)	10 (14.70%)	
Neurological severity of injury AIS (n%)	A	39 (32.23%)	3 (4.41%)	
B	43 (35.53%)	6 (8.82%)	
C	23 (19%)	21 (30.88%)	
D	16 (13.22%)	38 (55.88%)	
Paralysis type (n%)	Paraplegia	73 (60.33%)	56 (82.35%)	
Tetraplegia	48 (39.67%)	12 (17.65%)	
Cause of injury (n%)	Road traffic accidents	41 (33.88%)	–	
Falls (high/low)	38 (31.40%)	–	
Work-related injuries	20 (16.52%)	–	
Sports-related injuries	9 (7.43%)	–	
Other accidents#	13 (10.74%)	–	
Triple aortic aneurysms,	–	17 (25%)	
Transverse myelitis	–	14 (20.58%)	
Infections/Abscesses	–	11 (16.17%)	
Neoplasm	–	13 (19.11%)	
Viral infections	–	6 (8.82%)	
Canal stenosis	–	7 (10.29%)	
Age at the time of injury (years): mean, SD		36.9 (18.4)	58.3 (6.2)	
Age at the survey (years): mean, SD		38.1 (8.4)	60.6 (4.2)	
Duration since the injury (months)		15.2 (4.8)	27.4 (9.7)	
State of injury (n%)	Complete	22 (18.18%)	0 (100%)	
Incomplete	99 (81.81%)	68 (100%)	
Gender (n%)	Male	99 (81.82%)	50 (73.53%)	
Female	22 (18.18 %)	18 (26.47%)	
Age (years): mean, SD		34.6 ± 11.9	58.8 ± 13.2	
Marital status (n%)	Married	93 (82.60%)	68 (100%)	
Un married	28 (17.4%)	0 (100%)	
Level of education (n%)	Illiterate	5 (4.13%)	7 (10.29%)	
Elementary School	23 (19%)	16 (23.52%)	
Middle school	57 (47.10%)	34 (50%)	
High School	26 (21.48%)	8 (11.76%)	
College or more	10 (8.26%)	3 (4.41%)	
Employment status (n%)	Farmer	28 (23.14%)	6 (8.82%)	
Worker	36 (29.75%)	13 (19.11%)	
Government offices	17 (14.04%)	8 (11.76%)	
Retired	4 (3.30%)	35 (51.47%)	
Students	25 (20.66%)	3 (4.41%)	
Others*	11 (9.09%)	3 (4.41%)	
Notes:

Abbreviations: T-SCI, Traumatic Spinal Cord Injury; NT-SCI, Non-Traumatic Spinal Cord Injury; ASIA, American Spinal Injury Association scale; SCI, Spinal Cord Injury; C1–C4, Cervical 1-Cervical4; C5–C8, Cervical 5-Cervica8; T1–T6, Thoracic1-Thoracic6; T7–T12, Thoracic7-Thoracic12.

* included unemployed and self-employed individuals.

# included machinery-injury and violence injury.

Comparative analysis

According to the WHOQOL-BREF results, there were comparatively few significant differences in QoL domains and life satisfaction between the T-SCI and NT-SCI groups (p > 0.05), with the exception of the overall health domain, which showed a significant difference (p < 0.001; see Table 2). Furthermore, no significant differences were observed between the T-SCI and NT-SCI groups in the mean scores for physical health, psychological health, environmental health, and social relationships (p > 0.05). Both SCI groups exhibited mean scores lower than those of healthy Chinese individuals.

Table 2 Comparison quality of life domains and satisfaction with the life between T-SCI and NT-SCI.

Scale	Type of damage	
T-SCI
(n = 121)	NT-SCI
(n = 68)		SCI score	Chinese scores	
Mean (SD)	Mean (SD)	p-value	Mean (SD)	Mean (SD)	
SWLS	17.94 (5.79)	17.07 (5.05)	0.302	17.63 (5.54)	20.66 (3.57)	
Overall QoL	3.23 (0.99)	3.19 (0.82)	0.776	3.22 (0.93)	3.70 (0.50)	
Overall health	2.98 (0.90)	2.46 (0.87)	0.000a	2.79 (0.93)	3.39 (0.72)	
QoL-Physical health	48.32 (11.32)	49.06 (11.94)	0.674	48.59 (11.52)	59.99 (7.10)	
QoL-Psychological health	54.85 (10.78)	54.96 (9.24)	0.746	54.89 (10.23)	60.73 (8.64)	
QoL-Social relationship	2.46 (13.78)	50.38 (15.92)	0.348	51.71 (14.58)	65.24 (9.31)	
QoL-Environmental health	49.52 (13.19)	50.53 (13.39)	0.616	49.88 (13.24)	64.79 (6.12)	
Note:

Abbreviations: SWLS, Satisfaction with the Life Scale; NT-SCI, non-traumatic spinal cord injury; QOL, quality of life; SD, standard deviation; T-SCI, traumatic spinal cord injury.

a Significant results in bold, p < 0.05.

Figure 1 illustrates the rank order of the mean scores across all four domains for both T-SCI and NT-SCI participants. The highest scores were recorded in the psychological health domain (M = 54.85 for T-SCI and M = 54.96 for NT-SCI), while the lowest scores were found in the physical health domain (M = 48.32 for T-SCI and M = 49.06 for NT-SCI).

Figure 1 Rank order of WHOQOL-BREF domains scores of TSCI & NTSCI.

The satisfaction with life scale

The mean scores of the SWLS for both the T-SCI and NT-SCI groups showed no significant differences; however, both groups reported lower scores than healthy individuals (see Table 2).

Figure 2 illustrates the overall score distribution. The majority of participants in the T-SCI and NT-SCI groups reported being slightly dissatisfied, with percentages of 33.88% and 44.12%, respectively. This suggests that both groups have adapted to their physical disabilities. Additionally, dissatisfaction ranked as the second category in both T-SCI and NT-SCI groups, with 28.1% and 26.4%, respectively. The percentage of participants reporting extreme satisfaction was low, at 0.83% for the T-SCI group and 1.47% for the NT-SCI group. Conversely, 1.65% of participants in the T-SCI group and 1.47% in the NT-SCI group reported being extremely dissatisfied, while 0.83% of the T-SCI group fell into the neutral category.

Figure 2 Frequency distribution of SWLS.

Correlation analysis

Univariate analysis revealed moderate positive correlations between the outcomes. Notably, significant moderate correlations were observed between the SWLS and the QoL subscales, ranging from 0.09 (social dimension) to 0.32 (physical health dimension).

All QoL domains exhibited significant correlations among the physical, psychological, and environmental health domains (p < 0.01). Additionally, significant correlations were found between the psychological and environmental health domains, as well as between the social and environmental health domains (p < 0.01). The correlogram in Fig. 3 illustrates the correlations between the QoL and SWLS scores among all individuals with SCI across different domains.

Figure 3 Illustrates the correlations between the QoL and SWLS scores among all individuals with SCI across different domains.

Multiple linear regression analyses were conducted to determine the key factors influencing the WHOQOL domain scores, excluding items 1 and 2, with the SWLS score designated as the dependent variable. Variables were chosen based on their theoretical or clinical relevance, aimed at controlling for potential confounding factors, as well as their statistical significance. The four domains of QoL and SWLS scores were analyzed separately as dependent variables (see Table 3).

Table 3 Pearson’s correlation analysis among QoL and SWL of total people with SCI.

Characteristics	Physical health	Psychological health	Social	Environment	SWLS	
β	p-value	β	p-value	β	p-value	β	p-value	β	p-value	
Gender	3.31	0.103	0.85	0.667	14.67	<0.001	5.64	0.013	1.76	0.071	
Age	−5.38	<0.001	−1.51	0.177	1.18	0.393	−4.17	0.001	−0.93	0.09	
Marital status	3.07	0.044	1.88	0.203	1.28	0.486	1.32	0.437	1.37	0.06	
Level of education	0.12	0.929	−0.02	0.989	2.36	0.147	0.72	0.631	−0.32	0.615	
Employment status	−3.92	0.038	−1.31	0.473	−3.62	0.113	−5.50	0.009	−3.16	<0.001	
Duration since the injury	1.53	0.327	1.01	0.507	2.17	0.255	0.15	0.933	1.76	0.019	
Age at the time of injury	4.19	<0.001	0.30	0.799	−2.07	0.159	−0.37	0.784	1.08	0.064	
Type of injury	1.85	0.427	−0.29	0.898	−2.14	0.446	4.72	0.071	−2.13	0.056	
Level of injury	2.92	0.024	−1.11	0.376	0.85	0.586	2.32	0.109	2.30	<0.001	
Neurological severity of injury AIS	−0.42	0.695	−0.31	0.768	2.12	0.107	0.54	0.659	−1.01	0.053	
Paralysis type	7.42	0.044	−2.64	0.460	8.65	0.053	7.45	0.071	5.87	0.001	
Cause of injury	−1.25	0.245	0.83	0.425	0.68	0.598	−1.00	0.405	0.56	0.273	
State of injury	−3.14	0.312	5.20	0.087	−9.34	0.014	−1.44	0.679	2.06	0.166	
Note:

Abbreviations: SWLS, satisfaction with life scale; bold: significant value.

The analysis showed numerous significant predictors of physical health. Age, marital status, employment status, the level of injury and the type of paralysis were significantly associated to physical health domain of WHOQOL-BREF.

When examining psychological health, the analysis found that none of the predictor demonstrates significant relationships with psychological health.

Social health was positively influenced by gender. The type of paralysis had a marginally effect on social health, yielding a β of 8.65 (p = 0.053). However, the state of injury impacted social health, as evidenced by a β of −9.34 (p = 0.014).

In the context of environmental health, gender emerged as a significant predictor.

The analysis of SWLS revealed several important findings. Employment status, the level of injury and the duration since injury were associated with life satisfaction.

Discussion

This study compared the quality of life, health, and subjective well-being of individuals with T-SCI and NT-SCI. The results revealed significant differences in QoL and health between the two groups, particularly in the overall health domain as indicated by the WHOQOL-BREF scores. While self-reported SWB as measured by the SWLS was similar across both SCI groups, notable disparities emerged in the QoL and health domains, with the NT-SCI group reporting lower scores.

These findings align with previous research indicating that individuals with NT-SCI experience higher levels of physical, psychological, and environmental health (Migliorini, New & Tonge, 2011), while also reflecting lower scores in social relationships compared to their T-SCI counterparts (Barclay et al., 2019). The observed differences in QoL domains may be attributed to variations in the profiles of individuals with T-SCI and NT-SCI commonly referred to as a cohort effect. The overall health domain was significantly lower in the NT-SCI group compared to the T-SCI group. Notably, patients in the NT-SCI group were older at the time of onset, which may make them more susceptible to age-related comorbidities. Additionally, the strong relationship between health and QoL contributed to the lower overall QoL scores observed in the NT-SCI group compared to the T-SCI group.

The physical health domain of the WHOQOL-BREF assesses specific physical abilities. In this study, over half of the T-SCI patients (54.5%) had cervical injuries, compared to only 23.5% of those in the NT-SCI group. The T-SCI group had eight lumbar or sacral injuries, while the NT-SCI group had ten. Notably, the NT-SCI group had no complete injuries, whereas 22 patients in the T-SCI group did. This suggests that physical abilities are likely more restricted in individuals with T-SCI than in those with NT-SCI. Furthermore, the time since injury was shorter in the T-SCI group compared to the NT-SCI group. Ganesh & Mishra (2016) identified a strong relationship between post-injury duration and physical health-related quality of life, attributing this to a gradual reduction in pain and improved neurological function over time.

In the psychological domain, the current study found that the T-SCI group reported a mean score of 54.85 ± 10.78, while the NT-SCI group reported a mean score of 54.96 ± 9.24, indicating the presence of mood problems, anxiety, and depression. Some patients experienced difficulties with concentration, while others expressed dissatisfaction with their daily activities. These findings align with the study by Ganesh & Mishra (2016), which indicated a significant connection between psychological well-being and injury severity. Researchers attribute this connection to low self-esteem and negative feelings stemming from a perceived lack of control over their environment. Furthermore, self-efficacy and self-esteem are highly correlated with participation, contributing to a greater understanding of functioning, disability, and health (Geyh et al., 2012; Altahla et al., 2024; Altahla, Alshorman & Tao, 2024).

The social relationships domain scored higher in the T-SCI group compared to the NT-SCI group. Specifically, the T-SCI group’s score in the overall health domain was 2.98 ± 0.90, while the NT-SCI group’s score was 2.46 ± 0.87. This finding is consistent with previous research that indicated a strong relationship between social relationships and overall health (Hampton, 2004). Patients in the T-SCI group reported feeling more companionship, love, and support in their intimate relationships compared to those in the NT-SCI group. Qualitative studies have highlighted that individuals with NT-SCI often report feelings of being unsupported in managing their impairments (Guilcher et al., 2013; Barclay et al., 2017). These individuals tend to experience social isolation and rely heavily on family members for assistance in the absence of paid support (Barclay et al., 2017).

Conversely, the environmental domain scored higher in the NT-SCI group than in the T-SCI group. This is largely attributable to the fact that individuals with NT-SCI typically have incomplete injuries and lower injury levels. In this study, 100% of the NT-SCI group had incomplete injuries, and 61.9% had injuries at the T1–T12 level. As a result, the NT-SCI group demonstrated better physical health than the T-SCI group, which may explain their greater ability to navigate environmental barriers.

Hu et al. (2008) found significant differences in environmental factors between complete and incomplete SCI. Additionally, Boakye, Leigh & Skelly (2012) identified that environmental accessibility is related to neurological improvements, with higher accessibility in individuals with mild impairments compared to those with severe impairments.

Regarding the SWLS, life satisfaction levels were not significantly different between the SCI groups. This supports the findings of Barclay et al. (2019), which suggest that the etiology of the injury influences QoL outcomes following spinal cord injury. According to SWLS recommendations, an “average” score is between 20–24 and 15–19 is recommended as “below average” score (Pavot & Diener, 2008). It is common for SCI people to have low score with significant issues in many areas or to be doing well in many areas, but have one substantial problem.

A literature review examining mental health, life satisfaction, psychological factors, and social support has highlighted that individuals with SCI generally report lower life satisfaction compared to the healthy population. Previous studies often failed to differentiate between individuals with T-SCI and NT-SCI, with samples frequently being either mixed or composed solely of individuals with T-SCI (Post & van Leeuwen, 2012).

The current research found a positive correlation between SWB and QoL. This finding aligns with the study by Migliorini & Tonge (2009), which identified that the coping subscale of acceptance and the objective QoL domains were positively associated with normative SWB. Furthermore, a slight to medium positive relationship was observed between the physical, psychological, environmental health, and social relationships domains. This is consistent with findings from Ganesh & Mishra (2016). Additionally, the moderately significant association between SWB and physical health corroborates prior research (Fuhrer, 1994; Martin Ginis et al., 2010). However, only one study has specifically examined the relationship between physical activity and SWB in individuals with SCI (Ginis et al., 2003). Ginis et al. (2003) concluded that exercise can enhance life satisfaction and alleviate depressive symptoms, potentially through mechanisms related to pain reduction. In this study, the relationship between SWB and psychological health was found to be (r = 0.110, p = 0.133).

Evidence suggests that exercise can increase the production of neurotransmitters such as serotonin, norepinephrine, and dopamine, which play crucial roles in regulating emotions and preventing depression. While the relationship between SWB and the social domain was weak, this finding aligns with the study by Hampton & Qin-Hilliard (2004). In contrast, Fuhrer’s research strongly associated SWB with social relationships (Fuhrer, 1994). Additionally, SWB was significantly correlated with environmental health, suggesting that SWB is linked to the client’s social environment. Since SCI impacts the health of clients’ families, supportive family and community networks are vital for promoting SWB (Hampton, 2004).

Limitations

This study has several limitations. Firstly, the sample size was relatively small and was obtained through convenience sampling, which may limit the generalizability of the findings. Secondly, the cross-sectional design restricted participation to individuals undergoing rehabilitation following SCI, potentially overlooking the experiences of those outside this specific context. As a result, the findings may not be applicable to broader populations due to the limited sample size, specific demographics, and controlled conditions of the study.

To address these limitations, future research should aim to include a larger cohort of individuals with NT-SCI to enhance the robustness of the findings. Additionally, longitudinal studies conducted in the early stages following NT-SCI are needed to better evaluate QoL outcomes post-injury.

Conclusions

The findings indicate that individuals with NT-SCI face more marked challenges across various QoL domains, particularly in the overall health domain, with significantly lower satisfaction with life (SWL) scores compared to those with T-SCI. Factors beyond rehabilitation services and community support may also contribute to these disparities. Moreover, both SCI groups exhibited lower QoL and life satisfaction compared to the healthy population, underscoring the need for targeted interventions. The findings also highlight the complex interplay of various predictors on the QoL across different domains. Factors such as age, marital status, employment status, and type of paralysis play crucial roles in shaping physical, social, and environmental health. Physiotherapists and rehabilitation programs that work with NT-SCI patients should recognize that their QoL may be comparably lower than that of T-SCI individuals. It is essential for NT-SCI healthcare teams to focus on enhancing patient independence and facilitating community participation, as these efforts can lead to improvements in QoL. Importantly, a positive correlation was observed between SWB and QoL, suggesting that addressing both aspects is crucial for comprehensive rehabilitation strategies.

Supplemental Information

Supplemental Information 1 Raw data of 189 patients.

The authors would like to express their gratitude to the medical team of the Rehabilitation Department at Tongji Hospital for their valuable contributions and support throughout the study.

Additional Information and Declarations

Competing Interests

Author Contributions

Human Ethics

Data Availability

The authors declare that they have no competing interests.

Ruba Altahla performed the experiments, analyzed the data, prepared figures and/or tables, authored or reviewed drafts of the article, and approved the final draft.

Jamal Alshorman analyzed the data, authored or reviewed drafts of the article, and approved the final draft.

Sayed Zulfiqar Ali-Shah analyzed the data, authored or reviewed drafts of the article, and approved the final draft.

Mohammad Nasb analyzed the data, authored or reviewed drafts of the article, and approved the final draft.

Xu Tao conceived and designed the experiments, analyzed the data, authored or reviewed drafts of the article, and approved the final draft.

The following information was supplied relating to ethical approvals (i.e., approving body and any reference numbers):

The Tongji Medical College granted ethical approval to carry out the study within its facilities (approval number TJ-IRB20210314).

The following information was supplied regarding data availability:

The data is available in the Supplemental File.

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
