# Peer review of "Quality of life and subjective well-being comparison between traumatic, nontraumatic chronic spinal cord injury, and healthy individuals in China"

_PeerJ, doi:10.7717/peerj.18709_

## Round 0.1 · original submission · Major Revisions

Please improve the language and check the method for analysis. More statistical analysis methods should be explored.

Reviewer 1 ·

Basic reporting

The manuscript studies an interesting topic about the quality of life (QoL) and subjective well-being (SWB) comparison between traumatic (T-SCI) and nontraumatic spinal cord injury (NT-SCI) populations in China. This manuscript follows a conventional structure, which facilitates the comprehension of the study's aims, methods, results, and conclusions. However, the manuscript requires improvements in English language usage to enhance clarity and readability. The manuscript would benefit from a thorough review by a native English speaker to correct typos and improve clarity. You should also add more literature review regarding the status around the world. Below are some detailed comments:

Line 55: Please bold “Result”.
* Line 69-72: You should use newer references to provide more recent data.
* Line 72-73: Please provide a citation for this statement.
* Line 78: Please use the proper font for “(Nizeyimana et al., 2022) .”
* Line 79: there is a missing period between “Chang et al., 2020)” and “People”.
* Line 98: "comprised people with SCI" should be "comprised of people with SCI."
* Line 91: You should be specific about your wording. Is there no studies have been NT-SCI QoL or SWB around the world or in China?
* Line 103: "questionnaires were assigned by self-approach" is unclear. It would be clearer to say, "questionnaires were self-administered."
* Line 119: Please clarify how low the scores indicates poor QoL.
* Line 120-122: You mentioned “In this study”, are you sure you are mentioning this study? How would you get the two-week test and retest reliability with a cross-sectional study design like this study?
* Line 131: there is a missing ‘ ”. ’ between “good” and “31-35”.
* Line 181-182: "attributable to the overall health domain" is awkward. Consider rephrasing to "in the overall health domain."
* Line 265: You should mention the limitation is on the gereralizability of the study results.

Table 2. You should report the p values of 0.000 as “<0.001” throughout the manuscript. You do not a “a” for the “Significant results in bold”

Table 3. You should report the p values of 0.000 as “<0.001” throughout the manuscript.

Experimental design

While the statistical methods used (Student's t-test and Pearson's correlation) are generally appropriate, there are several critical issues that need to be addressed:

Normal Distribution Check: It is essential to check if the data follow a normal distribution to justify the use of Pearson's correlation. If the data are not normally distributed, Spearman's correlation should be used instead.

Confounding Factors: The use of Student's t-test does not account for potential confounding factors. Given the nature of the study, it is highly likely that the results are influenced by confounders. To address this, consider using multivariable regression analysis to adjust for possible confounders. Additionally, conducting several other regression analyses as sensitivity analyses would help support the robustness of the study findings.

Handling Missing Data: More detail should be provided on how missing data were handled. Specify any procedures or imputation methods used to address missing values in the dataset.

Validity of the findings

The results derived from the Student's t-test are highly likely to be biased by confounding effects. Without conducting multivariate regression analysis to adjust for these potential confounders, the validity of the study is questionable. To ensure the robustness of the findings, it is essential for the authors to perform multivariate regression analyses and related sensitivity analyses.

The study's correlation analysis only presented correlation coefficients that do not exceed 0.4, indicating negligible to weak correlations. This raises doubts about whether the study has demonstrated any meaningful correlations between the variables. However, it is important to note that the absence of strong correlations can also be meaningful, indicating that there may indeed be no significant relationship between the variables in this context.

·

Basic reporting

1. The title …… Chinese population
Do you mean the patients with TSCI and NTSCI in Chinese population. The original mean is likely the healthy Chinese population, but you do not recruit the healthy Chinese population. Please revise the title. If you recruit the Chinese healthy subjects, please revise the title “ …. among Traumatic, Nontraumatic , and healthy individuals in China”
2. Add the “chronic” word in your title to enhance your research motivation.
3. Line 130 ~ 133 Revise the sentence. what do you mean?
4. Line 155~156
“ few significant differences in QoL between two groups “ but you show the “P < 0.05” . It is a contradiction.

Experimental design

5. Methods section
Please offer the information about healthy Chinese population. Please add it. How many healthy Chinese healthy people do you recruit and its recruit criteria?

6. Table 2
“Chinese score” is strange description so it must be revised. Using the “Healthy subjects score” is appropriate. Other similar statements also need to be changed.
7. Line 160 ~ 162
please check the figure number and the figure title !!!!
8. Adjust the order of figures and table corresponding your manuscripts.
9. The y axis of figure 1 must be revised.

Minor concerns:
1. Line 73 ~75 Slightly revise the sentence.
“There could be differences in QoL and SWB between two groups, given …”
2. Line 75 ~ 78 Slightly revise the sentence of punctuation marks and ref format
3. Line 148 ~ 150, Grammatical errors
Of the patients, 121 (64.02%) had T-SCI, while 68 had NT-SCI, including 17 with triple aortic aneurysms and 14 with transverse myelitis…..
4. Line 151, Do not the omit the word “patients”
5. Line 230, check the ref
6. The P value in table 3 can be delete and use the “ * ” for presentation of significant.

Validity of the findings

Revise the results corresponding with your manuscripts

---

## Round 0.2 · Major Revisions

Please revise the manuscript according to the reviewer's comment.

Reviewer 1 ·

Basic reporting

Thank you for addressing all my previous comments regarding Basic reporting. I have no further comments.

Experimental design

No Comment.

Validity of the findings

Thank you for your response. I would like to emphasize the following points:

1. Use of t-test: While the Student’s t-test can indicate a significant difference between the two groups of SCI, this difference is highly likely to be influenced by confounding effects. For example, the difference in the health scores are very likely due to the difference in the mean age of two SCI groups (38.1 vs 60.6). You are basically comparing the health score of middle aged group with the elderly. Please reconsider your analysis to check what variables might introduce a confounding effect. Without adjusting for these potential confounders through multivariate regression analysis, the validity of the conclusions drawn from the t-test alone is questionable.

2. Correlation Analysis: A statistically significant correlation does not necessarily imply a meaningful relationship. The correlation coefficients presented in your study, which do not exceed 0.4, indicate negligible to weak correlations. This raises doubts about whether the study has demonstrated any meaningful correlations between the variables.

I recommend conducting multivariate regression analyses and related sensitivity analyses to ensure the robustness of your findings.

·

Basic reporting

OK

Experimental design

OK

Validity of the findings

OK

Additional comments

OK

---

## Round 0.3 · accepted · Accept

The comments have been addressed by the authors.

Reviewer 1 ·

Basic reporting

The authors have made significant strides in addressing the concerns raised in the previous review.

Experimental design

The authors have made significant strides in addressing the concerns raised in the previous review.

Validity of the findings

The authors have made significant strides in addressing the concerns raised in the previous review.